# Multiomics Molecular Research into the Recalcitrant and Orphan *Quercus ilex* Tree Species: Why, What for, and How

**DOI:** 10.3390/ijms23179980

**Published:** 2022-09-01

**Authors:** Ana María Maldonado-Alconada, María Ángeles Castillejo, María-Dolores Rey, Mónica Labella-Ortega, Marta Tienda-Parrilla, Tamara Hernández-Lao, Irene Honrubia-Gómez, Javier Ramírez-García, Víctor M. Guerrero-Sanchez, Cristina López-Hidalgo, Luis Valledor, Rafael M. Navarro-Cerrillo, Jesús V. Jorrin-Novo

**Affiliations:** 1Agroforestry and Plant Biochemistry, Proteomics and Systems Biology, Department of Biochemistry and Molecular Biology, University of Cordoba, UCO-CeiA3, 14014 Cordoba, Spain; 2Cardiovascular Proteomics Laboratory, Centro Nacional de Investigaciones Cardiovasculares (CNIC), 28029 Madrid, Spain; 3Plant Physiology, Department of Organisms and Systems Biology, University Institute of Biotechnology of Asturias (IUBA), University of Oviedo, 33006 Asturias, Spain; 4Evaluation and Restoration of Agronomic and Forest Systems ERSAF, Department of Forest Engineering, University of Córdoba, 14014 Cordoba, Spain

**Keywords:** *Quercus* spp., *Quercus ilex*, omics approaches, molecular markers, resilience, systems biology

## Abstract

The holm oak (*Quercus ilex* L.) is the dominant tree species of the Mediterranean forest and the Spanish agrosilvopastoral ecosystem, “dehesa.” It has been, since the prehistoric period, an important part of the Iberian population from a social, cultural, and religious point of view, providing an ample variety of goods and services, and forming the basis of the economy in rural areas. Currently, there is renewed interest in its use for dietary diversification and sustainable food production. It is part of cultural richness, both economically (tangible) and environmentally (intangible), and must be preserved for future generations. However, a worrisome degradation of the species and associated ecosystems is occurring, observed in an increase in tree decline and mortality, which requires urgent action. Breeding programs based on the selection of elite genotypes by molecular markers is the only plausible biotechnological approach. To this end, the authors’ group started, in 2004, a research line aimed at characterizing the molecular biology of *Q. ilex*. It has been a challenging task due to its biological characteristics (long life cycle, allogamous, high phenotypic variability) and recalcitrant nature. The biology of this species has been characterized following the central dogma of molecular biology using the omics cascade. Molecular responses to biotic and abiotic stresses, as well as seed maturation and germination, are the two main objectives of our research. The contributions of the group to the knowledge of the species at the level of DNA-based markers, genomics, epigenomics, transcriptomics, proteomics, and metabolomics are discussed here. Moreover, data are compared with those reported for *Quercus* spp. All omics data generated, and the genome of *Q. ilex* available, will be integrated with morphological and physiological data in the systems biology direction. Thus, we will propose possible molecular markers related to resilient and productive genotypes to be used in reforestation programs. In addition, possible markers related to the nutritional value of acorn and derivate products, as well as bioactive compounds (peptides and phenolics) and allergens, will be suggested. Subsequently, the selected molecular markers will be validated by both genome-wide association and functional genomic analyses.

## 1. Introduction

In 2004, Prof. Jesús V. Jorrín Novo (Agroforestry and Plant Biochemistry, Proteomics and Systems Biology Research Group) and Prof. Rafael M. Navarro-Cerrillo (Evaluation and Restoration of Agricultural and Forestry Systems Research Group), both at the University of Cordoba (Spain), started a pioneering joint project devoted to studying the molecular biology of the holm oak (*Quercus ilex* L.), as well as other forest tree species such as Pinus. This collaboration was born out of a double aim. Firstly, to understand different aspects of their biology, focusing on developmental processes (seed maturation and germination) and responses to environmental biotic and abiotic stresses using a molecular language throughout the backward flow of the central dogma of molecular biology, from phenotype towards genotype. Secondly, to decipher the molecular basis of the enormous phenotypic variability observed in *Q. ilex*, as well as the genetic structure of different populations correlated to edaphoclimatic conditions. However, we soon became aware of the project’s ambition, considering that the latest molecular techniques (e.g., modern omics techniques), used in biomedical research and, to a lesser extent, in plant model experimental systems such as *Arabidopsis thaliana*, were not being applied to this species. In general, the biological characteristics of forest trees have made them a challenging experimental system to work with, for example, due to their longevity, long regeneration period, allogamy, high phenotypic variability, and seed recalcitrance. For this reason, field experiments are difficult to carry out, making it necessary to include many individuals to reach consistent and reliable conclusions with biological significance [1]. The plant material selected for the experiments is young seedlings grown under greenhouse-controlled conditions, preferentially sampling leaves or roots. The experimental design must be scheduled on a yearly basis as, in the case of *Q. ilex*, seeds are collected in November–December. The recalcitrant character of the seeds makes it necessary to germinate them immediately after collection and perform the experiments by June–July without the possibility of repeating them until the next acorn harvest, one year later. Moreover, from our experience, standard and routine protocols and commercial kits successfully used in other plant systems, e.g., *A. thaliana*, do not work with *Q. ilex*, which requires an extra effort to optimize each step of the methodology (e.g., Echevarria-Zomeño et al. [2]).

The first pioneers’ biochemical papers on *Q. ilex* were published from the mid-1990s onwards [3,4,5]. From then until the 2020s, there were hardly any papers published on *Q. ilex*’s biochemistry and molecular biology, and the number devoted to the genus *Quercus* was quite low, which means that *Q. ilex* is still, like most *Quercus* spp., an orphan in molecular studies. Table 1 shows the number of publications found in the Web of Science (WOS) database at the time of writing this review, reporting on the use of omics approaches in *Quercus*, *Q. ilex,* and other organisms including several plant species and forest trees. The numbers illustrate that molecular research on forest tree species, especially *Q. ilex*, is minimal compared to that carried out with model and herbaceous plants, particularly compared to the number of papers on humans. Some reviews on forest tree species and *Quercus* have previously been published by the authors’ research group [6,7,8].

## 2. *Quercus ilex*, the Dominant Forest Tree Species in the Mediterranean Area (Why)

The genus *Quercus* (family Fagaceae, common name oak) emerged during global warming 55 million years ago [9], and is one of the largest genera of flowering woody plants, evergreen or deciduous, that produces acorn fruits; it dominates so-called oak forests and woodlands, natural or modified ecosystems, and constitutes a source of biodiversity [10]. The genus *Quercus* is one of the most relevant clades in the Northern Hemisphere in terms of number of species and ecological dominance [11]; in fact, the Plants of the World Online database [12] contains 460 *Quercus* spp. distributed throughout the Northern Hemisphere to Malaysia and Colombia.

Oaks have played an important role in human life since the prehistoric period and throughout different civilizations, being a source of a wide variety of goods and services, from wood for fuel and construction to food for human and animal consumption [13]. Despite being one of the most important woody species in terms of diversity, ecological dominance, and economic value [11], their molecular biology is scarcely known, and oaks can be considered, in general terms, an orphan of molecular studies, with some exceptions. Among those, it is worth mentioning *Quercus robur* L., considered the model within the genus, *Quercus lobata* Née, *Quercus suber* L. and *Q. ilex*, with this last species constituting our object of study for the past 20 years [8].

Within the genus, *Q. ilex* (Figure 1) is the dominant tree species in the Mediterranean forest and the agrosilvopastoral ecosystem “dehesa” (Spain) or “montado” (Portugal) (Appendix A) [14]. Hence, in Spain, out of the 15 Mha of forested area (50 Mha total Spanish land surface), 30% (4.5 Mha) corresponds to *Q. ilex*, mostly in “dehesas” forests (3,1 Mha; [15,16,17]). The most profitable product of *Q. ilex* is the acorns, which constitute the food of the Iberian pig, from whom high-quality meat products are created [18,19]. However, the true relevance of the *Q. ilex* plantations, not translatable into economic parameters, is related to environmental aspects such as carbon balance, water resources, biodiversity and, ultimately, desertification prevention [20].

Currently, there is a renewed interest in using orphan crops, including *Q. ilex* and other forest trees, for dietary diversification, sustainable food production and, ultimately, to improve nutrition, which is a challenge, especially in marginal areas [21,22,23]. *Q. ilex* fruits are currently undervalued and underexploited compared to other equivalent nuts such as the chestnut (*Castanea sativa* Mill.) and walnut (*Juglans regia* L.). Therefore, alternative economic uses are being considered for acorns, including as food [23], beverages [24], biofuel [25], pharmaceuticals, and in other industrial applications [26].

## 3. Biotechnology and Molecular-Assisted Breeding Programs for Resilience and Nutritional Purposes (What for)

Beyond the economic benefits, *Q. ilex*, like other forest tree species and their associated natural or transformed ecosystems, represent intangible wealth, a legacy we owe it to future generations to preserve. Consequently, forest management and conservation nowadays constitute a worldwide priority that should be reflected in international and national policies, integrating economic and environmental issues to generate richness and welfare [27]. The reality, however, is far from the theory, and a worrisome loss of forest spaces and increasing tree mortality has been occurring in recent decades [28], a problem that may worsen in a climate change scenario [29]. *Q. ilex* is a good example considering the increase in mortality observed over the last decade [30]. This is associated with anthropogenic factors and a combination of biotic and abiotic stresses. So-called oak decline syndrome is a result of combined stresses. In *Q. ilex*, the combination of drought episodes with the presence of the oomycete *Phytophthora cinnamomi* Rands is the main cause of the decline [31,32,33].

Faced with this concerning reality, it is imperative to orchestrate effective measures to ensure *Q. ilex* conservation, sustainable management, and successful reforestation to ultimately safeguard its ecosystem. Modern biotechnology, based on molecular knowledge, provides solutions that can be integrated with other chemical and biological alternatives [8]. The biological characteristics of *Q. ilex* preclude the biotechnological strategies successfully used in crops, such as genetic crosses, genetic engineering, and the search for agrochemicals. Thus, a unique feasible approach is based on the exploitation of biodiversity to select elite genotypes that will be used for clonal propagation later [34,35,36]. In fact, as discussed by Sork [37], the enormous phenotypic variability found in *Q. ilex* populations could be the result of its adaptation to environmental conditions, favoring the appearance of resilient phenotypes. It is, therefore, necessary to conduct the morphological, physiological, and molecular characterization of such phenotypic variability using molecular marker techniques. This will allow for the selection of elite genotypes in terms of higher germination rates and acorn production, desirable quality traits related to nutritional values, or adaptation and resilience to adverse biotic and abiotic stresses [7,8]. To date, the number of scientific works dealing with the molecular aspects of *Q. ilex* is rather limited due to its intrinsic difficulty as an experimental system (Table 1; [8]). This includes the high variability found in natural populations of *Q. ilex*, while *Populus* and *Eucalyptus* studies can use clones with sequenced genomes.

Despite the challenges it entails, we are addressing the molecular characterization of biodiversity in *Q. ilex* following the central dogma of molecular biology and using the omics cascade, which explains the flow of genetic information from DNA (genomics, epigenomics, and DNA-based markers) to metabolites (metabolomics), through mRNA (transcriptomics) and proteins (proteomics).

The use of state-of-the-art genomics and functional genomic approaches in model plants (e.g., *A. thaliana*) and crops (e.g., *Oryza sativa* L. and *Triticum aestivum* L.) has contributed significantly to our knowledge of the genetic underpinnings of phenotypes of interest, such as those related to acclimatation and adaptation to a changing environment, as well as to the selection of candidate marker genes to be used in breeding programs. In this direction, the research carried out by Prof. V. Sork on *Q. lobata* [9] and by Prof. C. Plomión in *Q. robur* [38] offers excellent roadmaps to follow in other *Quercus* spp., identifying geographic patterns of phenotypic and genotypic variations and performing complementary studies on pollen and seed movement, genomics, and epigenomics.

## 4. The “Omics” and Their Integration with Other Approaches (How)

During their 20 years of *Q. ilex* research, the authors have optimized many techniques and protocols, using both in silico and wet lab experiments, from classical biochemistry to the latest omics techniques. These customized procedures have been utilized in the characterization of mechanisms and the identification of genes and gene products associated with elite genotypes, more resilient to adverse environmental conditions (e.g., decline syndrome) and with higher nutritional value. Omics molecular data have been integrated with morphometric, physiological, and classical biochemical results, and with those of edaphoclimatic conditions, to correlate gene and gene products with specific phenotypes and environmental conditions, following a systems biology approach. In the case of acorns, metabolomics and proteomics data have been correlated with those of classical biochemistry (starch, sugars, amino acids, phenolics and flavonoids determination) and NIRS (near-infrared spectroscopy) [34,36,39]. The UHPLC-QToF (ultra-high performance liquid chromatography–quadrupole time-of-flight mass spectrometry) technique was used for untargeted metabolomic analysis. A total of 192 metabolites were annotated. Several acorn morphotypes were discriminated by principal component analysis (PCA), identifying 50 compounds out of 192 annotated with the highest load over the first two PCA components (explaining 67.2% of variability). These compounds could be considered potential markers of variability in *Q. ilex* acorns [36]. When determining the alimentary use of acorns, as nuts or for flour-derived products, sweetness is a desirable property [40] which is related to the water content and the chemical composition. The starch and sugar content is correlated to the sweetness, while tannins and other phenolics contribute to the astringent or bitter characteristics [41,42]. To date, we have not found a correlation between acorn size and chemical composition. Regarding seeds, they are of particular interest for studies aimed at unravelling the molecular bases of acorn recalcitrance and the differences between viable and non-viable seeds. In this matter, protein signatures for the different stages of the maturation and germination phases have been defined [43,44,45]. The interpretation of the omics data concerning the study of seedlings’ response to individual (drought or *P. cinnamomi*) or combined (drought and *P. cinnamomi*) stresses must take into consideration the physiological state of the seedlings sampled: mortality and damaged seedlings, water content and hydric regime, and photosynthesis-related parameters, as determined by a FluorPen FP100 portable leaf fluorimeter from Photon Systems Instruments (Drasiv, Czech Republic), a portable IR CO_2_ gas analyzer (IRGA) equipped with a light source integrated in a leaf chamber fluorometer and CO_2_ injector system (LiCor Li6400XT, LiCor, Inc.; Lincoln, NE, USA), and by pigment quantitation [33,35,46,47,48].

The following subsections are devoted to different DNA markers and omics techniques.

### 4.1. DNA Molecular Markers

A genetic marker is a sequence of DNA, present in specific genes, genotypes, populations, or species, that can be used in the identification of biological samples through hybridization, restriction mapping, or PCR techniques. Its high level of polymorphism, simple inheritance, and occurrence along the entire genome makes it preferable to morphological or biochemical markers. Practical applications in the plant field include studies of traceability, botanical systematics, genetic structure, divergence, phylogenetic relationships, and evolution. There are a wide variety of genetic markers that can be classified according to the technique they use and are known by the following acronyms: RFLP (restriction fragment length polymorphism), RAPD (random amplification of polymorphic DNA), AFLP (amplified fragment length polymorphism), SCAR (sequence characterized amplified region), SNP (single-nucleotide polymorphism), and SSR (single sequence repeat). Scotti-Saintagne et al. [49] compared different molecular markers (isozymes, AFLPs, SCARs, SSRs, and SNPs) to evaluate their usefulness for differentiating between two closely related oak species (*Q. robur* L. and *Quercus petraea* (Matt.) Liebl.).

DNA-based markers were employed in a study of a large number of species within the genus *Quercus*, as reviewed by Simeone et al. [50], Backs and Ashley [51], Lazic et al. [51], and Liang et al. [52], and are summarized in Appendix A. In general, data from DNA marker techniques clearly show a high level of genetic variation within the genus, especially in *Q. ilex* [53]. Several studies have reported on *Q. ilex* × *Q. suber* introgression events, more frequently in the direction *ilex* (female) × *suber* (male), which favors the immigration of *Q. suber* beyond its main range, in regions already colonized by *Q. ilex* [54]. Lopez de Heredia et al. [55]) analyzed, by ddRAD sequencing, adult hybrids (sensu lato) in a mixed stand and their open-pollinated progenies, estimating introgression levels in adults and seedlings, and concluded that the contribution of *Q. suber* to the genome is higher in progenies than in adults, which suggests preferential backcrossing with this parental species. Despite intraspecific ploidy variation within the genus *Quercus* being uncommon, Dzialuk et al. [56] have reported on the use of microsatellites to verify triploidy in *Q. petraea* and *Q. robur.* As an anecdotal curiosity, SSRs have been used in forensic botany, allowing for the identification of a leaf sample taken from a suspect’s vehicle as *Quercus and* placing the individual at or near a crime scene [57].

Among the different types of DNA markers available, our research group has used SSR markers for genetic structure and diversity studies in several populations of *Q. ilex* [58]. This type of marker is defined as having short tandemly repeated sequences, from one to six bases, distributed along the nuclear, mitochondrial, and chloroplast genomes [59]. Unlike other primer-based marker techniques, SSRs exploit the high level of homology occurring in the flanking regions of the SSRs among taxonomically related species. This feature enables cross-transferability—namely, markers developed for one species can be used for its relatives, which is especially important in species with little-studied genomes [60]. In fact, using SSRs, pure individuals from hybrids between *Q. ilex* and other species of the genus *Quercus* have been identified, such as *Q. ilex*–*Q. suber* [61,62,63] and *Q. ilex*–*Quercus coccifera* L. [64], and the analysis of the genetic structure and diversity of Spanish populations has been carried out [58,65]. We analyzed the genetic structure of several *Q. ilex* Andalusian populations using a battery of 10 chloroplast SSRs to examine maternal inheritance, and 10 nuclear SSRs to examine chromosomal recombination [58]. High levels of spatial differentiation were found for chloroplast DNA, indicating little seed dispersal between populations. As for nuclear SSRs, these exhibited much weaker spatial organization (differentiation between populations) than chloroplast DNA. The Betic Cordillera (Cádiz) population appeared to be consistently well separated from the northern Sierra Morena populations, suggesting that the Guadalquivir Valley has played an important role in determining population divergence (Appendix A). Currently, this battery of SSRs is being successfully used to characterize symptomatic and asymptomatic *Q. ilex* and *Q. suber* individuals located in areas of Spain affected by oak decline (Figure 2). So far, we have observed a possible differentiation between symptomatic and asymptomatic individuals analyzed in those areas (unpublished data). In a similar study developed in *Q. ilex*, PCR-RFLP and SSR markers show a clear geographic structure of genetic diversity [66].

### 4.2. Genomics

In the broadest and most general sense, genomics encompasses the structural and functional analysis of the complete genome of an organism, including mapping, sequencing, and gene expression profiling. At the time of writing, the number of land plant (Embryophyta) species whose reference genome has been made publicly available on GenBank was 897, with the figures for the Magnoliopsida (flowering plants) phylum, Fagales order, Fagaceae family, and *Quercus* spp. being 855, 20, 12, and 8, respectively. The value for forest tree species remains anecdotal compared with that of herbaceous models and crop species. By 2018, the first draft genomes of *Quercus* were released—those of *Q. robur* [38,67], and *Q. lobata* [9,68]— followed a few years later by *Q. suber* [69], *Q. mongolica* Fisch. ex Ledeb. [70], *Quercus aquifolioides* Rehder and E.H.Wilson, *Quercus wislizeni* A. DC., *Quercus gilva* Blume, and *Quercus glauca* (Thunb.) (Table 2).

As early as 1998, the genome sizes of four deciduous and three evergreen *Quercus* were reported, with values ranging from 1.88 to 2.00 pg/2C, revealing an average GC homogeneous content of 39.9%. In the case of *Q. petraea*, the DNA content differed between two populations, as confirmed by cytogenetic observations, displaying extra chromosomes in some individuals [71]. The characteristics and assembly statistics of the *Quercus* spp. whose genome has been sequenced are summarized in Table 2.

The achievement of genome sequencing and the analysis of gene sequences of forest species open the way for debate, speculation, and the formulation of hypotheses about the origin of mutations and evolutionary changes in long-lived organisms, how these contribute to coping with changes, and how to predict dynamics in a climate change scenario [38,72]. In this context, Sork et al. [9], while performing a high-quality de novo genome assembly of *Q. lobata*, traced its evolutionary history; interestingly, the analysis of 39,373 mapped protein-coding genes revealed duplications consistent with the genetic and phenotypic diversity underlying oak evolutionary success. Fitzek et al. [73] performed novel work with DNA barcoding to study introgression. By using restriction site-associated DNA sequencing, an SNP barcoding system was developed to characterize sympatric eastern North American white oak species and hybrids.

Our group is working on the first draft of the *Q. ilex* genome, using DNA extracted from leaves of an adult individual located in Aldea de Cuenca, Fuente Obejuna, Córdoba (coordinates: 38°3′95″ N; 5°55′40″ E) (Figure 1). For this purpose, the single-molecule real-time (SMRT) technology of the PacBio platform was used in collaboration with the University of Delaware. The genome dataset has been submitted to the NCBI BioProject repository (ID: 687489) and will be available soon. Previously, the genome size of *Q. ilex* was estimated by flow cytometry at 1.94 Gb/2C, and its 12 homologous chromosome pairs were identified using fluorescence microscopy (2n = 2x = 24) [6]. The preliminary assembled genome has a total length of 842 Mb.

### 4.3. Epigenomics

Epigenomics studies the stable inheritance of phenotypes resulting from changes in gene expression without alterations in the DNA sequence. Such changes are involved in the regulation of many biological processes and are mediated by different epigenetic mechanisms (DNA methylation, histone modification and micro-RNAs) [74]. To approach their study, cutting-edge platforms, techniques, and protocols have been continuously improved over the last decade [75]. Epigenomics is an emerging field of research in forest species that will increase our understanding of phenotypic plasticity and adaptative responses to environmental cues, including those related to climate change [76]. Despite scientific evidence supporting the involvement of epigenetic modifications in adaptation and responses to the environment, it is unknown to what extent genetic and epigenetic factors determine important traits related to fitness under these circumstances [77]. Such knowledge can be translated into biotechnology to address the worrisome increase in tree mortality observed in recent decades [78].

When we conducted a search on WOS with the keywords *Quercus* and epigenetics, 16 results were found (Appendix A). It is worth noting the work led by V. Sork’s group on *Q. lobata*, which can be considered a model system for the genus *Quercus* in epigenomics. They investigated how adaptive evolution occurs not only through genetic mutations but also through epigenetic mechanisms, such as DNA methylation [79]. Both genetic and epigenetic patterns showed local adaptation suggesting that CpG methyl polymorphisms are involved in the adaptation.

To our knowledge, only one paper, by Rico et al. [80], deals with the study of methylation marks in *Q. ilex* using the MSAP (methylation-sensitive amplification polymorphism) technique. The authors investigated the epigenetic modifications that may contribute to drought acclimation in natural forest trees, resulting in an increase in hypermethylated loci and a decrease in fully methylated plants exposed to drought. As a first approach to tackling epigenomics in *Q. ilex*, we are analyzing the global DNA methylation patterns at three developmental stages (leaves of adult trees and seedlings as well as embryos) from the *Q. ilex* individual used in whole-genome sequencing (Figure 1). We are employing a RAD-Seq variant and the MSAP-Seq (methylation-sensitive amplified polymorphism sequencing) technique, which has previously been described for *Hordeum vulgare* [81] and *Populus alba* [82] (Figure 3). Preliminarily, a higher degree of cytosine methylation was detected in adults (58%), followed by seedlings (33%) and embryos (9%) (unpublished data). Our results display a similar pattern to that reported for *Q. ilex* by Rico et al. [80], with 190 loci: 69% unmethylated, 22% full methylated, and 8% hemimethylated. Multivariate analyses conducted to explore the methylation status at MSAP loci also showed a clear distinction between developmental stages and organs. Regardless of the low number of loci commonly observed between developmental stages, some of them were present in trees and seedlings. To identify those putative differentially methylated genes expressed at the three developmental stages and two organs, the MSAP pre-PCR products obtained are being sequenced by the Illumina Inc. (San Diego, CA, USA) platform.

### 4.4. Transcriptomics

Transcriptomics deals with the study of the transcriptome, namely, the total set of RNA transcripts present in any cell or biological system at specific developmental and physiological stages and under specific environmental conditions. This definition reflects the dynamic character of the transcriptome, as changes in the RNA profiles determine the functional and phenotypic characteristics of the biological systems. The first reference transcriptomes for the genus *Quercus* were those of *Q. robur* and *Q. petraea*, obtained by Sanger and pyrosequencing technology [83]. Currently, there are more than 300,000 plant RNA sequencing data from organs, tissues, cells, developmental stages, and treatments available in different databases (reviewed by Lim et al. [84]). The platforms and methodologies used to obtain these data have evolved over time, from microarrays to what is now most common: RNA-seq [85,86]. In any case, these untargeted omics data must be validated by quantitative real-time PCR. However, data validation by quantitative real-time PCR data must consider the occurrence of alternative transcripts [87]. One of the key methodological issues at this point is the selection of suitable reference genes whose expression profile does not vary in response to the process under study [88]. In this context, we selected and evaluated 12 candidate reference genes not differentially expressed in *Q. ilex* seedlings in response to drought stress [89]. From the data obtained, and using the Bestkeeper, GeNorm, NormFinder and ΔCt comparative method algorithms, actin, GAPDH, and β-tubulin were selected as the most steady and reliable candidate reference genes (unpublished data).

When we conducted a search on WOS with the keywords of plants and transcriptomics, it resulted in 6680 items, with around 4% (238 items) corresponding to forest species. When the search was restricted to *Quercus*, a total of 28 items were displayed, with *Q. robur*, *Q. suber*, *Q. lobata,* and *Q. ilex* being the most represented species, agreeing with what was stated in the previous section and shown in Appendix A.

The priority in current research into *Quercus* spp. is the response to stresses and climate change conditions, and this has been approached using a comparative transcriptomic approach in different species [90,91,92,93]. A comparison of the least drought-tolerant deciduous *Q. robur*, quite tolerant deciduous *Q. pubescens*, and the most tolerant evergreen *Q. ilex* revealed common and species-specific responses [94]. Oney-Birol et al. [95] carried out a comparative transcriptome analysis in response to drought in different *Quercus* spp., identifying candidate genes that can contribute to adaptative divergence among hybridizing species. They found 398,042 variants across loci and 4532 variants in 139 candidate genes.

Transcriptomic analyses have also been very useful in shedding light on beneficial symbiotic and cooperative relationships, such as the interactions of *Quercus*–Mycorrhizas and *Quercus*–gall-forming insects, respectively [96,97]. The latter is an interesting example of cooperation between organisms, in which the gall-forming insects modify the plant’s genome to redirect the host plant tissue; there have been changes reported in 28% of the whole transcriptome between galled and ungalled tissue [98]. One of the changes observed in the inner larval capsule is the suppression of genes of the plant immune system by utilizing the host RNA interference mechanism. Furthermore, transcriptomics has been applied, together with DNA marker techniques (see Section 4.1), in the study of introgression between different *Quercus* spp. [95] and to unravel evolutionary aspects. Cokus et al. [99] developed a de novo assembled and annotated reference transcriptome for *Q. lobata* and *Quercus garryana* Douglas ex Hook, and identified SNPs, hypothesizing on the divergent selection for pathogen resistance election, which might explain ecological differences between species. Other aspects of marked practical interest studied through transcriptomics are *Q. suber* phellogen related to the quality of cork [100,101] and the molecular basis of seed recalcitrance [102].

The transcriptomic research carried out by our group in *Q. ilex* has mainly focused on the generation of RNA-Seq datasets to explore gene expression in several biological contexts, such as those related to environmental stresses and plant–pathogen interactions. The availability of such RNA-seq datasets has allowed for significant advances in the analysis of the stress responses in *Q. ilex*. The first transcriptome was generated from a mixture of tissues (leaf, root, and embryo) using the Illumina sequencing platform [103] and was further improved by a second one using the Ion Torrent platform [104]. A high number of transcripts—5405—were associated with stress response, of which 46 were involved in drought stress (Figure 4). Recently, the group has published a paper focused on the transcriptome (Illumina platform) and proteome (LC-MS/MS) profiles of six-month-old seedlings subjected to severe drought condition [89]. The quality and confidence of the mRNA and protein identifications and quantifications were assessed, identifying a total of 25,169 transcripts and 3312 proteins. Despite the poor correlation observed between mRNA and protein, four gene products (viz., FtSH6, CLPB1, CLPB3, and HSP22) were upregulated at both omics levels and, hence, were proposed as potential drought-tolerance markers to be used in the selection of elite, resilient genotypes and breeding programs.

Beyond the global RNA-seq analysis, the first transcriptomic analysis reported by our group dealt with the recalcitrant nature of *Q. ilex* acorns using a targeted transcriptomic approach combined with a hormonal and sugar analysis in mature seeds, germinated seeds, and seedlings, in an attempt to understand the physiological changes that take place during seed germination and seedling growth in *Q. ilex* [105]. Eleven genes were selected for the comparative analysis profile encoding proteins involved in desiccation tolerance (*DHN3* and *GOLS*), the regulation of ABA (abscisic acid) signaling (*OCP3*, *SKP1* and *SDIR1*), metabolism (*FDH*, *GADPH*, *RBLC* and *NADH6*), and oxidative stress (*SOD1* and *GST*). The transcript abundance of ABA-related genes (*OCP3*, *SKP1*, and *SDIR1*) was positively correlated with changes in phytohormone content (low levels of ABA were observed in mature seeds). Transcripts of drought-related genes (*DHN3* and *GOLS*), metabolism-related genes (*GAPDH* and *NADH6*) and oxidative stress-related genes (*SOD1* and *GST*) were more abundant in mature seeds. From our data it can be concluded that recalcitrance is established during seed development, which distinguishes between orthodox and recalcitrant seeds. In contrast, post-germination molecular events are quite similar for both types of seeds.

### 4.5. Proteomics

Proteomics is the scientific discipline whose object of study is the proteome, understood as the total set of proteins present in a biological unit (subcellular fraction, cell, tissue, organ, individual, ecosystem, or purified extracts) at a certain point in development, and under specific environmental conditions. Over time, an increasing number of proteomics branches have emerged, attempting to identify, catalog, and characterize a specific structural or functional group of proteins (proteases, phosphoproteins, membrane proteins, etc.) [106]. The field of plant proteomics was extensively reviewed from a conceptual, historical, methodological, and biological point of view by the authors’ group from 2003 [107] to 2021 [106]. Several of these studies discuss, in depth, the contribution of proteomics to plant biology [8,106,108,109,110,111], forest trees [6] or *Quercus* [7,8,112] knowledge. The field of proteomics, in relation to its application to the study of Mediterranean forest trees, was also reviewed by Pinheiro et al. [113]. Only three *Quercus* spp. have been examined using proteomics; *Q. robur* (1 publication by Sergeant et al. [114]), *Q. suber* (7 publications), and *Q. ilex* (20 publications).

The first report on the proteome in *Quercus* analyzed the protein profiles of *Q. suber* somatic and gametic in vitro culture-derived embryos using differential gel electrophoresis coupled with MALDI-TOF/TOF [115]. Subsequent papers dealt with somatic embryogenesis [116], responses to the oomycete *P. cinnamomi* [117], and cork-forming tissues, framed in a project aimed at studying cork quality [118,119].

The review by Rey et al. [7] discussed “How proteomics sees *Q. ilex.*” To some extent, the proteomes of individual organs (pollen, seeds, leaves, and roots) at different stages (mature trees and seedlings) under optimal and stress conditions have been characterized using classic 2-DE-based MS (number of identified proteins in a single experiment in the range of 400–600) and shotgun approaches (number of identified proteins in a single experiment in the range of 2000–4000 for data-dependent acquisition, and up to 6000 for data-independent acquisition). While the number of proteins confidently identified and quantified depends on the proteomic platform and database employed, our results indicate that the different platforms are complementary, providing new information, and hence allowing for deeper proteome coverage [8,120,121]. In this regard, the creation of a *Q. ilex* protein database based on available DNA/RNA sequences increased the quality (scores) and number of identified proteins in our experiments by almost three-fold [89,122].

By the mid-2000s, we had published the first proteome reference map of *Q. ilex* leaves, revealing the great plasticity of the organ, tree variability, and differences between developmental stages [123,124]. With the objective of characterizing *Q. ilex* populations, the high biological variability found in leaves prompted us to study acorns, which present a more stable proteome. Acorn 2-DE protein profile data allowed for the identification of group populations, correlating them with geographical location and climate conditions [34]. The same objective was pursued by analyzing the pollen proteome profile by 2-DE-MS and shotgun (LC-MSMS), corroborating the former data in terms of variability by provenance [125]. The effect and responses to stresses, drought and *P. cinnamomi* among individuals and populations were evaluated by analyzing the leaf proteome of 8–10-month-old seedlings. Echevarría-Zomeño et al. [46] reported that changes under drought and water-recovering conditions affected the proteins of photosynthesis, carbohydrate and nitrogen metabolism, as well as stress-related functional categories. A comparative proteomic analysis with two *Q. ilex* populations, differing in drought tolerance, revealed a general decrease in protein abundance, caused by drought, more pronounced in the less tolerant population and mainly affecting those involved in ATP synthesis and photosynthesis [48]. In parallel, these responses were also analyzed in the roots of three-month-old seedlings [126,127], with results pointing to metabolic adjustment (sugar mobilization and phenolic metabolism) and the activation of the defense system. Recently, we have gone one step further by proposing proteins and derived proteotypic peptides as markers of tolerance to drought to be used for the selection of elite tolerant genotypes in a breeding program. A panel of 30 proteins and 46 derived peptides was proposed, with two proteins (viz., subtilisin and chaperone GrpE protein) being upregulated under water stress in the three highly drought-tolerant populations studied [128].

*Q. ilex* responses to *P. cinnamomi* have been studied in one-year-old seedlings from two Andalusian provenances by analyzing changes in the leaf protein profile [47]. The general trend observed was a decrease in protein abundance upon inoculation, albeit slight in the least-susceptible population. The changes observed mostly affect chloroplast proteins involved in photosynthesis, the Calvin cycle, and carbohydrate metabolism, which agrees with what has been previously described in plants subject to drought. We have recently switched to studying combined stress responses in *Q. ilex*, recreating what occurs in nature and focusing on drought and *P. cinnamomi*, the two main causes of decline syndrome in Spain. San Eufrasio et al. [33] studied the effect and response of eight-month-old seedlings from three *Q. ilex* Andalusian populations—differing in their stress tolerance phenotype—to individual and combined stresses (*P. cinnamomi* and drought) using morphological, physiological, biochemical, and proteomic analyses. Photosynthetic proteins tended to be downregulated while those of stress-responsive proteins were upregulated. Although no treatment-specific response was observed for any functional group, protein changes corresponding to quantitative differences were mainly observed in response to the combined stresses. As a result, aldehyde dehydrogenase, glucose-6-phosphate isomerase, 50S ribosomal protein L5, and α-1,4-glucan-protein synthase (UDP-forming) are proposed as putative markers for resilience in *Q. ilex*.

Seed maturation and germination have also been studied in *Q.* ilex using a proteomic approach to understand the molecular basis of acorn recalcitrance. The protein profiles of the different parts of the seed, embryo, cotyledon, and teguments revealed a compartmentalization of metabolic pathways and a division of metabolic tasks [45]. Proteomic results and those from complementary approaches (gel-based and gel-free analyses) demonstrated that recalcitrance is established during seed development, while post-germination events were similar for both orthodox and recalcitrant seeds [44,121]. Our current research is evolving from a holistic to a targeted perspective. For example, Escandón et al. [129] have characterized several proteases and protease inhibitors in *Q. ilex* seeds by using proteomics combined with an in silico analysis of its transcriptome and in vitro and in gel activity assays. As mentioned above, proteases play a key role in stress responses and seed germination in *Q. ilex*, justifying the research focus on these enzyme families. Finally, proteomics generates knowledge with a clear translational potential in relation to the identification of allergens and bioactive peptides. Thus, Pedrosa et al. [130] characterized the first allergen reported in the pollen of *Q. ilex*, Que i 1. In a recent work carried out by our group (unpublished data), a total of 55 bioactive peptides encrypted in 2468 proteins were identified in the species-specific *Q. ilex* transcriptome database [89]. Of them, 12 bioactive peptides encrypted in 22 proteins were found in acorn protein extract by a shotgun analysis. The bioactive peptide NLAG was the most represented, being encrypted in seven tryptic peptides from four proteins, namely, enolase (qilexprot_67398), ketol–acid reductoisomerase (qilexprot_50910), 26S proteasome non-ATPase regulatory subunit 2 (qilexprot_34420) and Lysyl-tRNA synthetase (qilexprot_15726). It has been reported to be an inhibitor of the angiotensin-converting enzyme (ACE) and, therefore, to have antihypertensive activity [131].

### 4.6. Metabolomics

Metabolomics, the last omics to appear on the biological research scene, studies the metabolome, the total set of small molecules (Mr lower than 1500 Da) of a biological system—from subcellular fractions to whole organisms—as the product or intermediate of different metabolic pathways. From six papers published by 2001, the number increased to 4562 by 2021 (search on WOS). Of the 14,464 items found on WOS, 4302 corresponded to plants, 95 to forest species, 13 to *Quercus*, and 9 to *Q. ilex*. The lowest analytical potential of metabolomic techniques, resulting in hundreds of metabolites identified in a single experiment, compared to tens of thousands of transcripts and thousands of proteins identified in equivalent transcriptomics and proteomic experiments, is counterbalanced by being the closest to the phenotype in the flux of genetic information.

The metabolome is highly complex and dynamic and includes a wide range of physicochemical heterogeneous molecules from primary and secondary metabolic pathways, representing a major challenge for the analytical tools employed. These include classical biochemistry procedures based on colorimetric reactions and non-destructive NIRS, along with mass spectrometry and NMR (nuclear magnetic resonance) platforms for untargeted global analysis. As with the other omics, metabolomics requires the use of algorithms and databases for metabolite annotation, quantitation, and statistical analysis [132].

The field of plant metabolomics, and specifically forest tree metabolomics, has been subject to periodic review [133,134,135,136,137]. Metabolomics, either alone or in combination with other omics techniques, has generated new knowledge on the molecular basis of biological processes in systems biology, such as responses to stresses in herbaceous plants and crops and, to a lesser extent in forest trees [138,139]. Within the genus *Quercus*, metabolomic studies have been reported in *Q. robur*, *Q. suber*, *Quercus pubescens* Mills, *Q, petraea*, *Quercus aliena* Blume, *Quercus agrifolia* Née, and *Q. ilex*. Differences in the metabolite profile between species, populations, and individuals have been studied to understand the ecological and evolutionary role of phytochemicals [140]. Buche et al. [141], using untargeted LC-MS, described relevant compounds of *Q. robur* and *Q. petraea* wood samples, proposing the use of metabolite signatures as markers of wood quality. In the same direction, the phenolic composition of phellem and its relationship with wood quality has been evaluated in *Q. suber* [119]. In *Q. robur*, differences in primary and secondary metabolites between insect-resistant and -susceptible individuals were found. While tolerant trees were enriched in secondary metabolites (tannins, flavonoids, and terpenoids), susceptible ones were enriched in carbohydrate and amino acid derivatives, supporting the hypothesis of a constitutive defense against insects [142,143].

A few studies have dealt with responses to drought and climate change conditions. In a four-year *Q. pubescens* drought trial, GC-MS metabolomics revealed that tolerance implicates the tricarboxylic acid cycle shunt through the glyoxylate pathway, and the galactose metabolism by reducing carbohydrate storage and increasing proteolytic activity [92]. Changes in the phenolic pattern in *Q. suber* during drought and recovery treatments have been reported [144], and the process of metabolite resorption from plant leaves during senescence has been evaluated under drought and warming conditions in *Q. rubra* [145].

Concerning metabolomic studies in response to pathogens, seasonal variations in the phloem phenolic profile and in response to *Phytophthora ramorum* Werres, De Cock and Man in ‘t Veld were evaluated in *Q. agrifolia* [146], finding a correlation between the phenolic profile and resistance level. A topic of great interest, but scarcely represented in the literature, is the truffle–*Quercus* interaction. In this regard, Li et al. [147] investigated the *Tuber indicum* P. Micheli ex F. H. Wigg–*Q. aliena* symbiosis at the metabolomic level.

With respect to *Q. ilex*, Prof. J. Peñuelas’ group evaluated seasonal-, climate change-, and drought-induced metabolic changes through field experiments, with a special emphasis on volatiles [148,149]. Observed changes under moderate drought conditions included a higher content of sugars and phenolics, with these changes being related to folivory [150]. Drought tolerance is linked to signaling and osmoregulation by hexoses, and to antioxidant protectors such as polyols and amino acids [151]. The work carried out by Rivas-Ubach et al. [152] revealed differences in the metabolite profile depending on the soil characteristics and tree size.

Our group approached *Q. ilex* metabolomics with different objectives using complementary platforms, including classical biochemical colorimetric assays, NIRS, and untargeted metabolomics by GC-MS and LC-MS platforms. The first published work in this field [34] reported on natural variability in *Q. ilex* Andalusian populations based on acorn morphometry and chemical composition determined by NIRS analysis. Two main clusters were detected, corresponding to the northern and southern provenances, with the former having higher protein, sugar, linoleic, and oleic acid content. Because of the renewed interest in the alimentary use of acorns, either as nuts or for flour-derived products, the phytochemical composition variability among morphotypes was determined by LC-MS [36]. Of the 3259 (positive mode) and 1841 (negative mode) resolved compounds, 192 metabolites were annotated, with differences mostly in the carbohydrates, amino acids, and secondary metabolites. Some of the identified compounds had bioactivity (antimicrobial, insecticide, allelochemical, antioxidant, antitumor, proangiogenic, etc.), proving the nutraceutical value of the acorns. The most recent work published by our group focused on the response to drought stress in *Q. ilex* seedlings by using a non-targeted metabolomic approach [153]. The leaf metabolome has been described to determine possible mechanisms and molecular markers of drought tolerance, as well as identify putative bioactive compounds. A total of 3934 compounds were identified: 616 variable and 342 annotated (Figure 5). Of the annotated compounds, 33 were variable, mostly corresponding to amino acids, carboxylic acids, benzenoids, flavonoids, and isoprenoids.

By using a multiomics approach, including transcriptomics (NGS-Illumina), proteomics (shotgun LC-MS/MS), and metabolomics (GC-MS), *Q. ilex* metabolism has been partially reconstructed [154]. These analyses resulted in the identification of 62,629 transcripts, 2380 protein species, and 62 metabolites. Of the 127 metabolic pathways reported in the KEGG pathway database, 123 can be visualized using the described methodology. The TCA cycle was the pathway most represented, with 5 out of 10 metabolites, 6 out of 8 protein enzymes, and 8 out of 8 enzyme transcripts.

Ongoing research is addressing changes in the metabolite profile in response to individual or combined (drought and *P. cinnamomi*) stresses, and the search for resilience metabolite markers; furthermore, we are evaluating metabolic changes during seed germination.

## 5. Conclusions and Future Directions

In 2004, our group started a research project on molecular biology with forest tree species, mainly focused on *Q. ilex*. The general objective was to understand the biology of the species, follow the central dogma of molecular science using the omics cascade, and integrate the omics data obtained with those from morphology, physiology, and classical biochemistry in the direction of systems biology. For this, our specific objectives were to characterize variability, genes, and gene products implicated in the response to biotic and abiotic stresses related to decline syndrome and climate change, later evolving towards studies of seed recalcitrance and ultimately to acorn phytochemical analysis, related to its nutritional and nutraceutical value as seeds and flour-derived products. The knowledge generated is intended to be translated into breeding programs, assisted by molecular markers and based on the selection of resilient and productive elite genotypes, and into seed conservation strategies to aid the search for bioactive compounds. Nowadays, we are embracing new objectives, such as the analysis of plant biostimulants [155,156] and the priming phenomenon [157] for protecting *Q. ilex* against stresses related to decline syndrome. Preliminary unpublished results show that seeds treated with BTH (benzothiadiazole), a chemical analogue of the defense-related molecule salicylic acid, display an increased and accelerated germination rate, reducing seedling infection by *P. cinnamomi*.

This project is proving challenging due to the recalcitrance of the species as an experimental system, the enormous intra- and interpopulation variability found, and characteristics such as its long life cycle and allogamy. Field data are difficult to interpret in biological terms, and greenhouse-controlled experiments are limited to a short period of time each year. Beyond doubt, the use of clonal individuals generated by vegetative propagation or somatic embryogenesis will favor molecular research and data analysis [158,159]. The allogamous character of *Q. ilex* makes it extremely difficult to draw conclusions about inheritance, since acorns from individual trees, and the seedlings and plants derived from them, are highly variable due to the unknown origin of the pollen. Reliable interpretations of the results would only be possible by including a high number of biological replicates in the experiment, which is incompatible with omics studies. In this regard, it would be convenient to conduct the experiments under controlled pollination conditions, although no optimized protocols for this have been reported in the literature so far. As an example, in a recent experiment to assess the response to the combined stress of drought and *P. cinnamomi* in the offspring of symptomatic and asymptomatic *Q. ilex* trees surveyed in areas of decline, all possible combinations, with symptomatic and asymptomatic seedlings coming from symptomatic and asymptomatic mother trees, were seen (unpublished data). This clearly illustrates the complexity of the resilient phenotype in which genetic, epigenetic, and environmental factors are involved, and which remains poorly investigated in healthy or diseased individuals of *Q. ilex*, in particular, and of the genus *Quercus* in general.

However, our current research with *Q. ilex,* together with that reported on other *Quercus* spp., mainly *Q. lobata*, *Q. robur*, and *Q. suber*, confirm that the most recent and cutting-edge molecular biology techniques successfully used with model plants and crops are suitable for this species. We intend, following the path of other *Quercus* spp. research (e.g., https://quercusportal.pierroton.inra.fr/; https://corkoakdb.org/ accessed on 19 July 2022), to update the *Q. ilex* database portal (holm oak database at http://www.uco.es/investiga/grupos/probiveag/ accessed on 19 July 2022); the whole omics dataset will be deposited and available to the scientific community. At the methodological level, we will continue with our holistic strategy to deepen and expand the current coverage of the genome, transcriptome, proteome, and metabolome, while moving towards a directed approach. We do not disregard classical biochemical techniques in our analysis: even the simplest colorimetric ones provide relevant information for the selection of specific genotypes for further studies using more complex and costly approaches [7]. To gain deeper insight into the reality of *Q. ilex* biology and avoid analytical bias, the whole omics dataset, with thousands of data generated from wet experiments, must be integrated with morphological, biochemical, and phenotypic data from the systems biology direction. For that, a substantial effort is necessary in bioinformatics to generate biological models using tools such as DIABLO that are very useful in this direction [160]. A modest and preliminary approach to this strategy was the network reconstruction of metabolic pathways in *Q. ilex* [154]. However, such an objective is challenging since the number of molecular entities identified by each omics layer is different at scales of tens of thousands, thousands, and hundreds, for transcriptomics, proteomics, and metabolomics, respectively. In general, we have not found good correlations between transcripts and protein levels, due to either analytical or biological reasons. For example, in our experiments on stress responses, the number of upregulated gene products visualized at the mRNA level, concurrent with the proteins they encode, occurring twice and in different individuals, are usually very low [89].

The sequencing of the *Q. ilex* genome opens many possibilities for future research, of which we will prioritize the implementation of state-of-the-art approaches such as a genome-wide association study (GWAS) [161], as well as the identification of SNP variants [162]. Epigenetics is a priority in our current and future research [76]. We have just started to address this field by measuring the global methylation level of cytosines at different development stages and organs using the MSAP technique, and have concluded that it is not a very informative technique for this purpose. These results have determined the switch to bisulfite sequencing to characterize DNA methylation markers, in addition to expanding our studies of epigenetic mechanisms to the modification of histones and small RNAs, mostly linked to priming and stress memory [163]. The availability of genomic, transcriptomic, and proteomic information in our database will greatly facilitate the identification of pivotal enzymes in epigenetic mechanisms.

Finally, it is worth mentioning our intention to open up a new line of research, that of functional or reverse genomics [164], to validate those genes and gene products proposed as candidate markers of resilience [33,35,89]. This task is unachievable within *Q. ilex*, since the genetic resources and protocols for transformation and regeneration are not available. For this, it is necessary the use heterologous functional validation systems, such as *A. thaliana* or *Nicotiana benthamiana,* as has already been reported for *Q. suber* genes related to floral organ identity and drought response [165,166].

In summary, the objectives that we pursued with our research on *Q. ilex* were previously proposed by Plomión et al. [167] and achieved to a certain extent with other forest tree species [168].

## Figures and Tables

**Figure 1 ijms-23-09980-f001:**
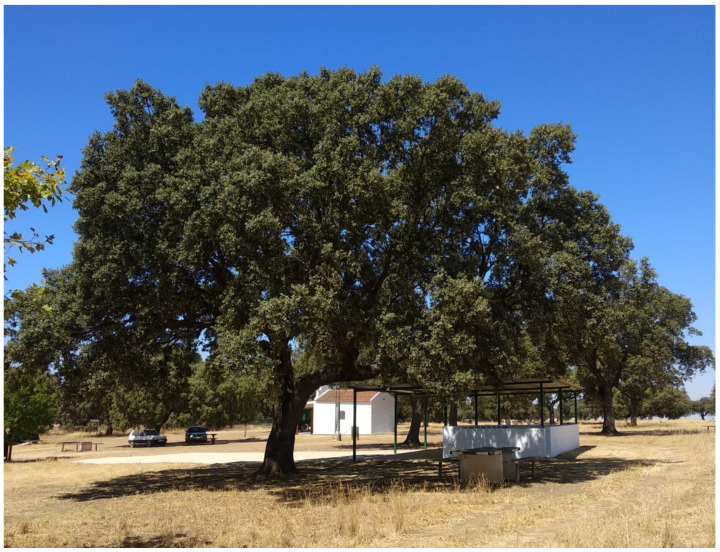
Tree used to obtain the first draft genome of *Q. ilex*, located in Aldea de Cuenca, Fuente Obejuna, Cordoba, Andalusia, Spain (38°19′46″ N, 5°33′15″ W).

**Figure 2 ijms-23-09980-f002:**
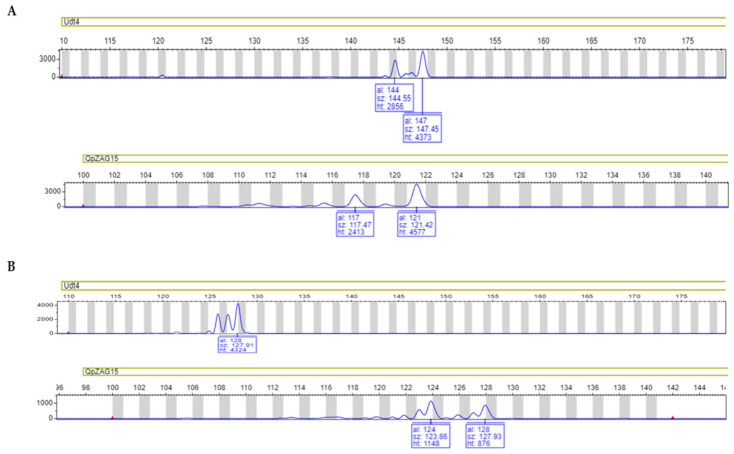
Electropherogram of Udt4 (chloroplast) and QrZAG15 (nuclear) SSR locus in *Q. ilex* (**A**) and *Q. suber* (**B**).

**Figure 3 ijms-23-09980-f003:**
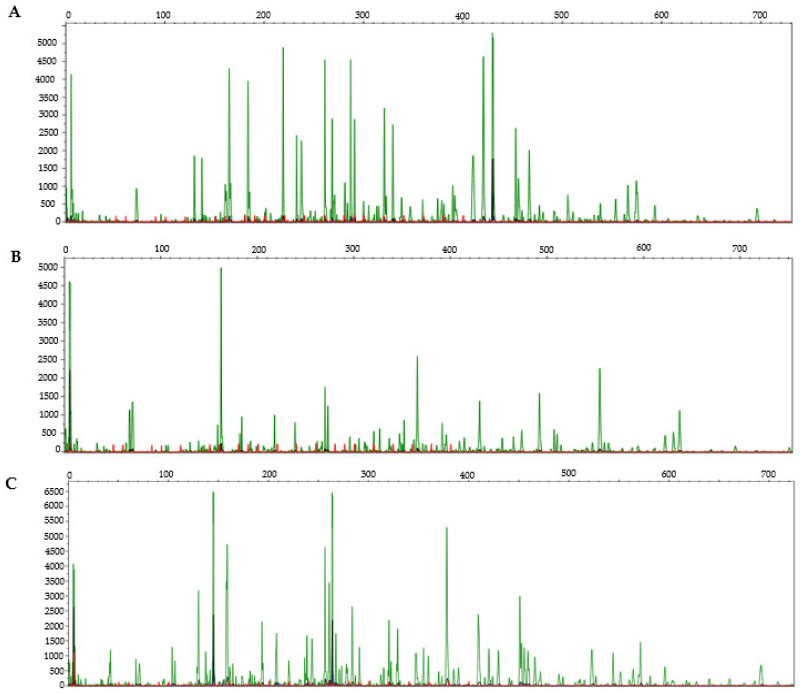
Examples of MSAP polymorphisms in electropherograms corresponding to *Q. ilex* adult tree leaves (**A**), embryos (**B**) and seedling leaves (**C**) using MspI + EcoRI digestion and the primer combination EcoRI + ACG and MspI + TTA.

**Figure 4 ijms-23-09980-f004:**
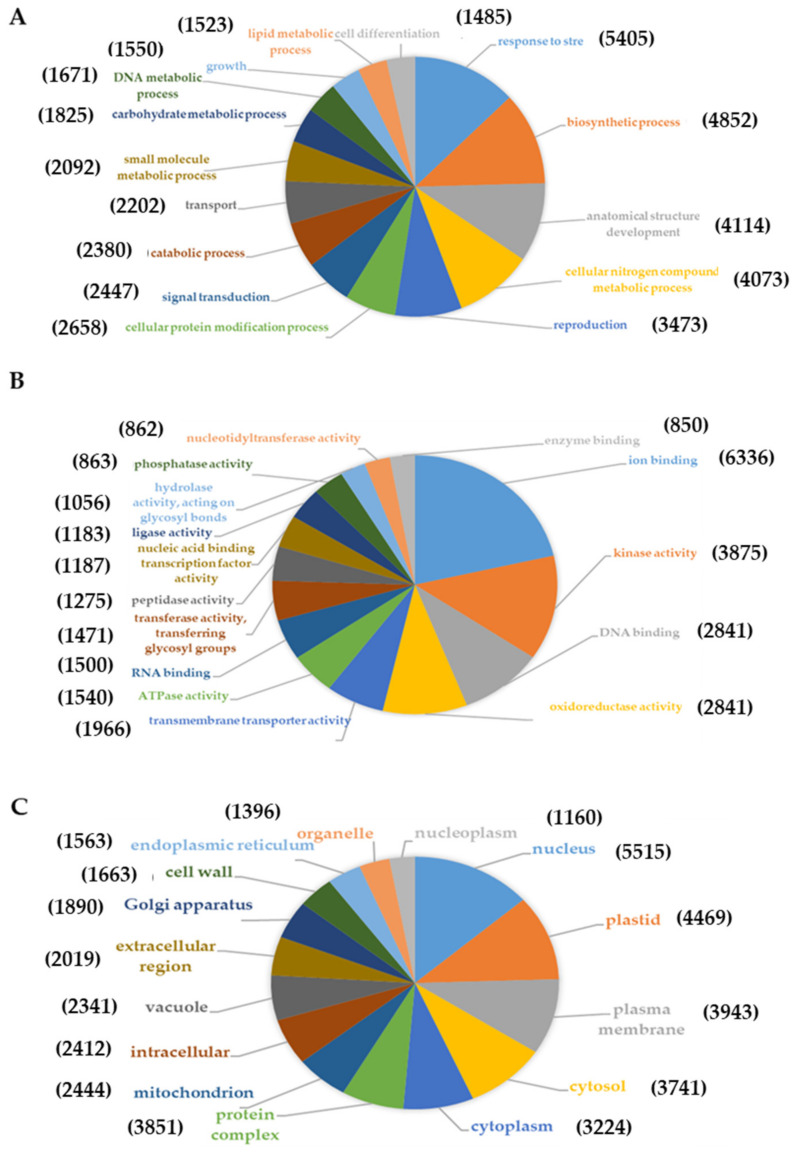
Pie charts of GO classification of assembled *Q. ilex* transcripts including the first 15 transcripts assigned to biological processes (**A**), molecular functions (**B**) and cellular components (**C**). The number in parentheses indicates the number of transcripts. The remaining transcripts assigned to each GO category are shown in Guerrero-Sánchez et al. [104].

**Figure 5 ijms-23-09980-f005:**
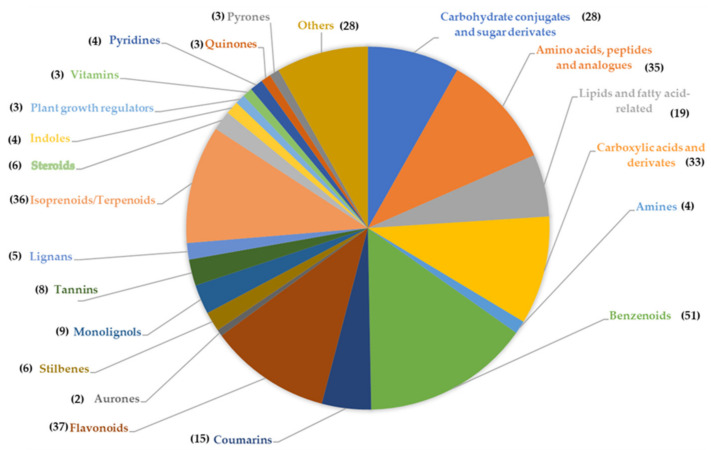
Pie chart of the chemical families identified in *Q. ilex* seedlings subjected to drought stress. The number in parentheses indicates the annotated compounds [148].

**Table 1 ijms-23-09980-t001:** Number of papers found in the Web of Science (WOS) database (28 July 2022) from a search using the name of the organism (first column) and the omics approach employed (second to fifth columns) as keywords (included in the abstract).

Organisms	Genomics	Transcriptomics	Proteomics	Metabolomics
*Quercus*	27	9	20	9
*Quercus ilex*	0	3	15	4
*Populus*	161	29	34	28
*Pinus*	70	11	28	12
*Eucalyptus*	55	8	26	14
*Arabidopsis*	1197	293	723	344
*Rice*	1137	153	452	312
*Saccharomyces*	548	101	387	192
Humans	11,064	2842	11,458	4975

**Table 2 ijms-23-09980-t002:** Assembly statistics of the whole-genome sequencing of *Quercus* spp.

*Quercus* spp.	Genome Size (Mb)	Contig N50 (Mb)	Number of Scaffolds	GC Content (%)	Repetitive Sequences (%)	Protein-Coding Genes	NCBI Genbank ^b^
*Q. robur* ^a^	814.3	1.35	1409	35.5	53.30	25,808	GCA_932294415 [38,63]
*Q. lobata*	845.9	0.97	2002	35.0	54.00	39,373	GCA_001633185 [8,64]
*Q. suber*	953.3	0.81	23,347	36.0	11.96	33,658	GCA_002906115 [65]
*Q. mongolica*	809.8	2.40	321	35.5	53.75	36,553	GCA_011696235 [66]
*Q. aquifoloides*	926.5	1.40	212	36.5	NA ^c^	NA	GCA_019022515
*Q. wislizeni*	724.8	9.00	358	35.0	NA	NA	GCA_023055345
*Q. gilva*	889.8	28.30	515	35.5	NA	NA	GCA_023621385
*Q. glauca*	903.1	7.60	415	35.5	NA	NA	GCA_023736055

^a^ The haploid genome of *Q. robur*. ^b^ NCBI Genbank (National Center for Biotechnology Information Genbank database). ^c^ NA indicates data have not been published yet. This information has been obtained from the NCBI Genbank.

## Data Availability

Not applicable.

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
