# Peer review of "Multiomics Molecular Research into the Recalcitrant and Orphan Quercus ilex Tree Species: Why, What for, and How"

_ijms, 2022, doi:10.3390/ijms23179980_

Round 1

Reviewer 1 Report

This manuscript outlines the importance of molecular data for Quercus ilex and the various techniques that have been applied to deliver those data (as stated in the title: why, what for and how). The review seems to aim to emphasize the history of the application by the authors’ group of DNA-based markers, genomics, epigenomics, transcriptomics, proteomics, and metabolomics for research on Quercus ilex. As a reader I am more interested in knowing more about the results, knowing what is unique to forest trees, what to Quercus, and what to Quercus ilex in particular. Information about this is presented but not always in an easily accessible way. This is done to a larger extend in the 2021 review by a subset of the authors, here quoted as reference 7: Molecular research on stress responses in Quercus spp.: From classical biochemistry to Systems Biology through Omics analysis. Maybe this is why a different approach was taken this time, but I question whether the current manuscript is not better placed in more specialized journal instead of a more general journal like ijms with an impact factor of 6.2.

Questions that came up while reading this manuscript include:

Line 75 “plant systems” – what plant systems? The herbaceous Arabidopsis, or the woody Populus and Eucalyptus?

Line 86 “It is important to mention that some reviews on forest tree species and Quercus, have been published by the authors´ research group [5-7].” – why is this important to mention?

Line 91 why indicate the number of publications by the author’s group?

Line 648 and further: Are the various challenges unique to Quercus ilex or also present in other Quercus spp., or in Populus or Eucalyptus?

Line 162 Table1 shows how many papers have been published on various organisms. This could be a supplementary Table in my view.

Line 206 “with references to our current research and publications and those found by other authors in Quercus spp.” Does not need to be stated. Similar to the sentence in line 447/448.

Related to this, I suggest to clarify which species were examined in the referenced articles. For example, mention the tree species in line 262, lines 334/335, line 338.

It would be desirable, in my view, to be more specific as to what the findings were of the quoted articles. What data did the research bring? This information is presented at times but could be made clear in a concluding statement of each section and/or be included in the supplementary the tables.  Related to this, can anything be said based on what is known to date with the different techniques about similarities or differences in how Quercus and the more extensively studied Populus react at the molecular level to environmental conditions- do they likely have similar or different strategies? Does it seem that one technique yields more useful information compared to another technique? This can be especially important given the often poor correlation between eg transcriptome and proteome data.

To clarify what I mean I give some examples here:

Line 187 “In the case of acorns, metabolomics and proteomics data have been correlated with those of classical biochemistry (starch, sugars, amino acids, phenolics and flavonoids determination) and NIRS (near infrared spectroscopy) trying to associate differences in molecular composition with sweetness or bitterness and fruit morphotype [33, 35, 37].” Which compounds correlated with sweetness?

Line 323 “revealing that large portions of the genome are involved [75] where CpG DNA methylation marks, but not CHG, seems to play an important role.” This is quite vague. A question that comes up in my mind, for example, is whether this study suggest that environmental conditions cause epigenetic changes that are maintained and lead to better survival? And is this finding supported by the Rico et al [76] paper?

Line 359 I suggest to include a remark that the interpretation of quantitative real-time PCR to validate RNAseq data must take into account the occurrence of alternative transcripts (one of the possible reasons that transcript and protein data often do not correlate, as is mentioned in lines 410 and 690).

Line 712 “as have been already reported in Q. suber genes related to floral organ identity and drought response [160-161].” Apparently, the function of some Q. suber genes have been verified – are these genes popping up in the transcriptome or proteome data for Q. ilex?

The Supplementary Tables present a list of publications using certain techniques for different objectives but falls short of informing the reader whether the results helped answer questions. It would help to organize the data by technique or by objective.

Table 2: State what NA(c) stands for

Figure 5 is lacking, except for the legend.

Grammar and spelling:

In general, the authors have the tendency to use very long sentences. I suggest breaking up most of these. I list here some suggestions for text changes but note that the complete text should be checked because there are many more places where improvements can be made:

Put all Latin names of species in italics (unless the journal advices otherwise)

Line 38 Replace “pretend” with “intend”

Line 39 Replace “Systems direction proposing possible molecular markers related to resilient and productive genotypes to be used in reforestation programs, and to nutritional value of acorn and derivate products, searching bioactive compounds (peptides, phenolics) and allergens.”  I suggest dividing the sentence into two or three.

Line 42 Replace “the molecular markers selected” with “the selected molecular markers”

Line 52 Replace “molecular bases” with “molecular biology”

Line 57 Replace “molecular bases” with “molecular basis”

Line 71 Replace “makes necessary germinate” with “makes it necessary to germinate”

Line 73 Replace “of repeat them” with “of repeating them”

              Replace “one year ahead” with “one year later”

Line 85 Replace “at a sidereal” with “at a considerable”

Lines 140/141 Rewrite sentence

Line 146 Replace”, successful reforestation, and ultimately safeguards” with “, and successful reforestation, to ultimately safeguard”

Line 149-153 Break up and rewrite the sentence. I do not quite understand what the authors wish to covey here.

Line 193 Replace “aimed to unravel” with “aimed at unravelling”

Author Response

Reviewer 1: Comments and Suggestions for Authors

This manuscript outlines the importance of molecular data for Quercus ilex and the various techniques that have been applied to deliver those data (as stated in the title: why, what for and how). The review seems to aim to emphasize the history of the application by the authors’ group of DNA-based markers, genomics, epigenomics, transcriptomics, proteomics, and metabolomics for research on Quercus ilex. As a reader I am more interested in knowing more about the results, knowing what is unique to forest trees, what to Quercus, and what to Quercus ilex, in particular. Information about this is presented but not always in an easily accessible way. This is done to a larger extend in the 2021 review by a subset of the authors, here quoted as reference 7: Molecular research on stress responses in Quercus spp.: From classical biochemistry to Systems Biology through Omics analysis.

Answer: The previous mentioned review is focused on stress responses and gives a general overview of the studies performed with different Quercus species. The present one is focused on Q. ilex, with some comparisons with other Quercus spp, and mostly based on the research carried out by the authors group in the almost last 20 years. It includes not only responses to stresses but other basic studies such as those on seed maturation and germination. We have also added some discussion on translational issues as the alimentary use of acorns and flour derived products. The message we intend to transmit is the need of doing basic molecular research aiming at developing strategies for Q. ilex and related natural and agroforestry ecosystems management, conservation, and exploitation, in the direction of the sustainable development strategies. This point is being discussed along the review. As the referee says, and indicated in the title, the questions why, what for and how have directed the writing of the review. In this new version of the manuscript, we have introduced changes, as suggested by the referee, to discuss more in detail some results. In any case, a deeper discussion can be found in the original papers published so far, and because of that an extensive reference list was included. We are aware that Q. ilex, as experimental system, can be of limited interest, but is the system we are been working with. The molecular route to perform studies with this species, and probably other orphan studies, is well drafted, and this is one of the main objectives of the review. Even so, we are at the beginning of knowing and understanding the biology of the species. Once the “wet and dry” approaches have been settled, we will focus on biological questions and hypothesis. By now, our main interest is related to the search of genes and gene products that can be used as markers of interesting phenotypes (resilience and productivity), markers that can be used in breeding programs. The validation and identification of those genes (e.g. SNPs) will be carried out in the next step.

Maybe this is why a different approach was taken this time, but I question whether the current manuscript is not better placed in more specialized journal instead of a more general journal like ijms with an impact factor of 6.2.

Answer: Our point of view is different, although we respect the referee opinion. From a methodological point of view, the review fits perfectly in the scopus of the journal, although, we can discuss about the interest of a minor forest tree species as Q. ilex, in particular, and Quercus spp., in general.

We have previously published three papers in IJMS, and, in total, at least 13 on different Quercus spp. have appeared in different IJMS issues. Probably, the referee did not consider that the present review corresponds to a Special Issue, "State-of-the-Art Molecular Plant Biology Research in Spain". The corresponding authors are guest editors of the SI. As we are a Spanish group, we have oriented our manuscript to show the “state of the art” of a “Spanish” species but with a “global view”, so, we thought in submitting this review. The title and general idea were communicated to IJMS, being accepted. They encouraged us to the submission.

Questions that came up while reading this manuscript include:

Line 75 “plant systems” – what plant systems? The herbaceous Arabidopsis, or the woody Populus and Eucalyptus?

Answer: This has been clarified in the manuscript.  “Besides, from our experience, standard and routine protocols and commercial kits successfully used in other plant systems, e.g., A. thaliana, does not work with Q. ilex, which means an extra effort to optimize each step of the methodology accordingly. As an example, it has been shown in Echevarria-Zomeño et al. [2].”

Line 86 “It is important to mention that some reviews on forest tree species and Quercus, have been published by the authors´ research group [5-7].” – why is this important to mention?

Answer: We have rewritten the sentence, eliminating “is important to mention” The previous reviews were focused on just proteomics or the more recent one on stress responses in the genus Quercus. “Some reviews on forest tree species and Quercus, have been previously published by the authors´ research group [6-8].” Ref. 5 and 6 (now ref. 6 and 7) deal with the contribution of the approach to the study of forest tree species (Ref. 6) and Q. ilex (Ref. 7), being this topic updated in the proteomics specific section.

Line 91 why indicate the number of publications by the author’s group?

Answer: I agree, it is not relevant, and has been removed in the revised version.

Line 648 and further: Are the various challenges unique to Quercus ilex or also present in other Quercus spp., or in Populus or Eucalyptus?

Answer: We have worked with other Quercus spp. and they are also difficult species to work with, at least, from a molecular point of view. We have never worked with other forests species such as Populus or Eucalyptus. The difficulty of working with Q. ilex is mainly related to the high variability found on natural populations, which makes more difficult data interpretation. On the contrary, in the case of Populus or Eucalyptus, the use of clones makes easier the reproducibility of results as well as data interpretation. At the same time, the availability of their sequenced genomes facilitates molecular studies. Moreover, the economic interest of Populus or Eucalyptus justifies the differences in the number of research group and publication leading these species. 

Line 162 Table1 shows how many papers have been published on various organisms. This could be a supplementary Table in my view.

Answer: This table pretend to show, based on data, the situation of the omics research in Q. ilex, and the comparison with other Quercus, forest trees, herbaceous spp., Saccharomyces, and humans. From our point of view, the table is illustrative, and simple, so we think is better to keep it. Other more complex Tables are included as supplementary material.

Line 206 “with references to our current research and publications and those found by other authors in Quercus spp.” Does not need to be stated. Similar to the sentence in line 447/448.

Answer: It has been deleted in the revised version.

Related to this, I suggest clarifying which species were examined in the referenced articles. For example, mention the tree species in line 262, lines 334/335, line 338.

Answer: This information has been included in the manuscript: “We are employing a RAD-Seq variant, the MSAP-Seq (Methylation Sensitive Amplified Polymorphism-Sequencing) technique, which has been previously described in Hordeum vulgare [81] and Populus alba [82] (Figure 3). Preliminarily, a higher degree of cytosine methylation was detected in adults (58 %), followed by seedlings (33 %) and embryos (9 %) (unpublished data). Our results display a similar pattern to that reported in Q. ilex by Rico et al. [80].”

It would be desirable, in my view, to be more specific as to what the findings were of the quoted articles. What data did the research bring? This information is presented at times but could be made clear in a concluding statement of each section and/or be included in the supplementary the tables. Related to this, can anything be said based on what is known to date with the different techniques about similarities or differences in how Quercus and the more extensively studied Populus react at the molecular level to environmental conditions- do they likely have similar or different strategies? Does it seem that one technique yields more useful information compared to another technique? This can be especially important given the often-poor correlation between e.g., transcriptome and proteome data.

Answer: We agree with the referee´s comment, but we did not pretend to describe and discuss every aspect of a quite ample topic, either from a methodological or biological point of view, but just focusing on the data and discussion of our own research. The readers have the references to go through if they are interested. However, as the referee suggests, we will include the main conclusion and findings of the papers reported in the supplementary tables. Each issue raised by the referee could be the subject of a review. As for example, the poor correlation between mRNA and protein abundances is quite common and has been recurrently discussed in Guerrero-Sánchez et al., 2021 Journal of Proteomics. There are a high number of analytical or biological reasons, whose discussion exceed the objective of the review. We have commented on this in some other previous reviews just to point at these issues.

To clarify what I mean I give some examples here:

Line 187 “In the case of acorns, metabolomics and proteomics data have been correlated with those of classical biochemistry (starch, sugars, amino acids, phenolics and flavonoids determination) and NIRS (near infrared spectroscopy) trying to associate differences in molecular composition with sweetness or bitterness and fruit morphotype [33, 35, 37].” Which compounds correlated with sweetness?

Answer: The paragraph has been rewritten as follow: “In the case of acorns, metabolomics and proteomics data have been correlated with those of classical biochemistry (starch, sugars, amino acids, phenolics and flavonoids determination) and NIRS (near infrared spectroscopy) (33, 35, 37; now 34, 36, 39). The UHPLC-QToF (ultra-high performance liquid chromatography-quadrupole time-of-flight mass spec-trometry) technique was used for untargeted metabolomic analysis. A total of 192 metabolites were annotated. Several acorn morphotypes were discriminated by a PCA (Principal Component Analysis) analysis, identified 50 compounds out for 192 annotated with the highest load over the first two PCA components (explaining 67.2% variability). These compounds could be considered as potential markers of variability in Q. ilex acorns [36]. While pretending the alimentary use of acorns, as nuts or flour derived products, sweetness is a desirable property [40], which is related to the water content and the chemical composition. Starch and sugar content is correlated to the sweet property, while tannins and other phenolics contribute to the astringent or bitter characteristic [41-42].”

Line 323 “revealing that large portions of the genome are involved [75] where CpG DNA methylation marks, but not CHG, seems to play an important role.” This is quite vague. A question that comes up in my mind, for example, is whether this study suggest that environmental conditions cause epigenetic changes that are maintained and lead to better survival? And is this finding supported by the Rico et al [76] paper?

Answer: The information has been clarified in the manuscript: “They have investigated how adaptive evolution occurs not only through genetic mutations but also through epigenetic mechanisms, such as DNA methylation [79]. Both genetic and epigenetic patterns showed local adaptation suggesting that CpG methyl polymorphisms are involved in the adaptation.” Reference 75, now 79, describes an intimate connection between CpG methylation and adaptative evolution. Also, they report that the CHG methylation is not directly related to adaptative evolution. However, no information about whether the CpG methylation are maintained and lead to better survival. In this regard, Rico et al. [80] reported an experiment with natural Q. ilex populations to assess whether the epigenetic modifications allow developing a rapid acclimation response to stress conditions. For this, they determine the level of DNA methylation in individuals in either unstressed or stressed plots subjected to drought for 12 years. They reported that the hypermethylated loci (CmCmGG/GGCmCm) increased and the fully methylated loci (CCmGG/GGCmC) decreased in individuals under drought conditions without evidence about the maintenance of epigenetic marks. As the long-term treatment developed, these epigenetic modifications were linked to a lessening of the drought effect. They concluded that Q. ilex shows a large capacity to rapidly acclimate to changing environmental conditions; however, they also do not give information about whether these epigenetic marks are maintained (We do not currently know which characteristics of DNA methylation prevail in natural populations or what their impacts on evolutionary processes might be).

Line 359 I suggest including a remark that the interpretation of quantitative real-time PCR to validate RNAseq data must take into account the occurrence of alternative transcripts (one of the possible reasons that transcript and protein data often do not correlate, as is mentioned in lines 410 and 690).

Answer: We have incorporated in the revised version the following sentence: “However, data validation by quantitative real-time PCR to validate RNAseq data must consider the occurrence of alternative transcripts [87].”

Line 712 “as have been already reported in Q. suber genes related to floral organ identity and drought response [160-161].” Apparently, the function of some Q. suber genes have been verified – are these genes popping up in the transcriptome or proteome data for Q. ilex?

Answer: Reference 161, now 166, reports on changes in the transcript profile in Q. suber roots from plants subjected to drought stress, with 546 differentially expressed unigenes. Despite being different organs, leaf and root, some of the genes, such as chaperons, LEA, dehydrins, among others, also popped up in the transcriptome and proteome data in Q. ilex. Regarding reference 160 now 165, we have not performed floral development studies. It was included to illustrate some work on data validation.

The Supplementary Tables present a list of publications using certain techniques for different objectives but falls short of informing the reader whether the results helped answer questions. It would help to organize the data by technique or by objective.

Answer: We have modified supplementary tables included a new column with main conclusions of answers to the questions.

Table 2: State what NA(c) stands for

Answer: This information has been clarified in the manuscript: “NA indicates data has not been published yet. This information has been obtained from NCBI Genbank.”

Figure 5 is lacking, except for the legend.

Answer: This figure has been included in the manuscript.

Grammar and spelling:

In general, the authors have the tendency to use very long sentences. I suggest breaking up most of these.

Answer: Some of long sentences have been broken up.

I list here some suggestions for text changes but note that the complete text should be checked because there are many more places where improvements can be made:

Put all Latin names of species in italics (unless the journal advice otherwise)

Answer: All of them has been written in italics.

Line 38 Replace “pretend” with “intend”

Answer: Done

Line 39 Replace “Systems direction proposing possible molecular markers related to resilient and productive genotypes to be used in reforestation programs, and to nutritional value of acorn and derivate products, searching bioactive compounds (peptides, phenolics) and allergens.”  I suggest dividing the sentence into two or three.

Answer: Done

Line 42 Replace “the molecular markers selected” with “the selected molecular markers”

Answer: Done

Line 52 Replace “molecular bases” with “molecular biology”

Answer: Done

Line 57 Replace “molecular bases” with “molecular basis”

Answer: Done

Line 71 Replace “makes necessary germinate” with “makes it necessary to germinate”

Answer: Done

Line 73 Replace “of repeat them” with “of repeating them”

              Replace “one year ahead” with “one year later”

Answer: Done

Line 85 Replace “at a sidereal” with “at a considerable

Answer: Done

Lines 140/141 Rewrite sentence

Answer: Done

Line 146 Replace”, successful reforestation, and ultimately safeguards” with “, and successful reforestation, to ultimately safeguard”

Answer: Done

Line 149-153 Break up and rewrite the sentence. I do not quite understand what the authors wish to covey here.

Answer: These sentences have been rewritten to improve clarity. “Q. ilex is a good example of this situation considering the increase of mortality observed in the last decade [30]. This is associated to anthropogenic factors and a combination of biotic and abiotic stresses. Within the combined stresses, it is remarkable the so-called oak decline syndrome. In Q. ilex, the combination of drought episodes with the presence of the oomycete Phytophthora cinnamomi Rands. are the main causes of the decline syndrome [31-33].”

Line 193 Replace “aimed to unravel” with “aimed at unravelling”

Answer: Done

Reviewer 2 Report

This review on multi-omics research of the recalcitrant and orphan Quercus ilex tree species is well written and interesting. It provides a comprehensive overview of genomic, transcriptomic, proteomic, and metabolomic studies that have been conducted on the orphan tree species Quercus ilex. However, I do have minor comments and suggestions:

- Regarding the title, it is redundant to say multi-omics molecular research because omics include already molecular.

- It would be great to have a distribution map of Quercus ilex species in the world as well.

Author Response

Reviewer 2: Comments and Suggestions for Authors

This review on multi-omics research of the recalcitrant and orphan Quercus ilex tree species is well written and interesting. It provides a comprehensive overview of genomic, transcriptomic, proteomic, and metabolomic studies that have been conducted on the orphan tree species Quercus ilex. However, I do have minor comments and suggestions:

Regarding the title, it is redundant to say multi-omics molecular research because omics include already molecular.

Answer: We appreciate the reviewer´s recommendation, but we would like to keep the word "molecular" in the title even though it sounds redundant.

It would be great to have a distribution map of Quercus ilex species in the world as well.

Answer: The following Figure S1 about the distribution map of Q. ilex in the Mediterranean forest and the agrosilvopastoral ecosystem “dehesa” (Spain) adapted from [169].

Round 2

Reviewer 1 Report

I am happy to see the extensive editing of the text, thank you. This certainly helps the reader to better understand what points the authors wish to make.  I have some further text suggestions at the end of this report for consideration.

I indeed had missed that this article is to be part of the Special Issue, "State-of-the-Art Molecular Plant Biology Research in Spain". My apologies. I did not want to argue that an article about Quercus ilex is misplaced in IJMS, I apparently did not make myself clear on that, rather that such article should appeal to a broader audience such as researchers working on poplar or eucalyptus, who are interested in learning about Q. ilex research and how that compares to what is known about their favorite plant. In other words, that it would be good to compare the information on Quercus ilex research with that on other Quercus species and similar plants such as poplar and eucalyptus, to give the reader a better appreciation of where the current Q. ilex research stands. Re-reading the revised manuscript in the context of the Special Issue convinces me that this is done sufficiently.

Line 286. It would be great if it could be clear where the Guadalquivir Valley is in relation to the Betic Cordillera and Northern Sierra Morena populations. Can these (and other?) regions be indicated in Figure S1? Did the nuclear markers reveal anything?

Line 472/473/474. Please rephrase this sentence as it is now not clear. Maybe DELETE “, when the patterns are similar for orthodox and recalcitrant seeds”?

Supplementary Tables: I appreciate the work the authors have undertaken to include results. I am sure future readers will appreciate this as well!

Figure S1: I suggest adding the names of the various regions, especially those that are mentioned in the text.

Text change suggestions:

Line 120/121: Please clarify what is said here, I seem to misinterpret this sentence. 4.5 MHa is more than 20% of 15 MHa.

Line 156: REPLACE “in the main cause” WITH “is the main cause”

Line 175: REPLACE “system (Table 1[8]).” WITH “system (Table 1[8]). This includes the high variability found in natural populations of Q. ilex, while Populus and Eucalyptus studies can use clones with sequenced genomes.”  (NOTE that I used here the comment you gave in your response).

 Line 182: DELETE “relevant”

Line 232: REPLACE “and -omics techniques are presented.” WITH “and -omics techniques.”

Line 240: DELETE “, for use as an isozyme”

Line 249/250: REPLACE “exploring the distribution of differentiation at the genome level in” WITH “differentiating between” [Check if this still captures what you want to say here]

Line 256: REPLACE “, more noticeable in” WITH “, especially in”

 Line 264: REPLACE “neing” with “being”                                                                                                                                       

Line 282: REPLACE “10 chloroplasts that” WITH “10 chloroplast SSRs that”

Line 282: REPLACE “that indicate maternal” WITH “ to examine maternal”

Line 283: REPLACE “that indicate chromosomal” WITH “to examine chromosomal”

Line 284: REPLACE “found with” WITH “found for”

Line 370: REPLACE “described in” WITH “ described for”

Line 373: REPLACE “reported in” WITH “reported for”

Line 374: REPLACE “with 190 loci represented, 69 %” WITH “for 190 loci: 69%”

Line 399: REPLACE “For this, one” WITH “One”

Line 411: DELETE “is”

Line 412: “is the responses to”  EITHER “is the response to” OR “are the responses to”

Line 414: DELETE “the”

Line 426: REPLACE “in to mRNA profile of 28 % of” WITH “to 28% of”

Line 472: REPLACE “concluded” with “indicate”

Line 488: REPLACE “approached by using” WITH “examined by”

Line 512: italicize “Q. ilex”

Line 525/526 REWRITE TO IMPROVE CLARITY please

Line 532: REPLACE “of potential use in breeding programs based on the selection of elite tolerant genotypes.” WITH “to be used for the selection of elite tolerant genotypes in a breeding program.”

Line 553 etc: REPLACE “As a result, the following variable proteins are proposed as putative markers for resilience in Q. ilex, namely, aldehyde dehydrogenase, glucose-6-phosphate isomerase, 50S ribosomal protein L5, and α-1,4-glucan-protein synthase (UDP-forming).” WITH “As a result, aldehyde dehydrogenase, glucose-6-phosphate isomerase, 50S ribosomal protein L5, and α-1,4-glucan-protein synthase (UDP-forming) are proposed as putative markers for resilience in Q. ilex.”

Line 603: REPLACE “is generating” WITH “has generated”

Line 723: REPLACE “Research” WITH “ research”

Line 765: REPLACE “Benthamiana” with “benthamiana”

Line 766: REPLACE “in” with “for”

Author Response

Comments and Suggestions for Authors

I am happy to see the extensive editing of the text, thank you. This certainly helps the reader to better understand what points the authors wish to make.  I have some further text suggestions at the end of this report for consideration.

I indeed had missed that this article is to be part of the Special Issue, "State-of-the-Art Molecular Plant Biology Research in Spain". My apologies. I did not want to argue that an article about Quercus ilex is misplaced in IJMS, I apparently did not make myself clear on that, rather that such article should appeal to a broader audience such as researchers working on poplar or eucalyptus, who are interested in learning about Q. ilex research and how that compares to what is known about their favorite plant. In other words, that it would be good to compare the information on Quercus ilex research with that on other Quercus species and similar plants such as poplar and eucalyptus, to give the reader a better appreciation of where the current Q. ilex research stands. Re-reading the revised manuscript in the context of the Special Issue convinces me that this is done sufficiently.

Dear colleague,

First of all, thank you very much for your very professional and deep revision and edition of the manuscript. Your contribution, comments and criticisms, allowed us to improve it a lot. Thanks also for your understanding. The focus of the paper was determined by the special issue it will be included. Without such a SI we should not be confident in preparing such a review. The molecular biology and knowledge of Q. ilex is very much behind that of other Quercus (lobata and robur) and other forest tree spp. (populus, eucalyptus, and even some pines). It is to early to get a clear idea on similitudes and differences. We have designed the methodological route, and from now we can get into biology with more confidence. To us, and by now, what is important is to understand the high variability within Q. ilex, and the differences between populations, individuals, and even between the descendants of individual trees. Apart of its recalcitrance as experimental system, the high variability found impedes to compare with other species. We are publishing soon the first draft of the species and from that we can perform wide genomic associations based on SNPs, and this will be the starting point to make confident conclusions. 

Line 286. It would be great if it could be clear where the Guadalquivir Valley is in relation to the Betic Cordillera and Northern Sierra Morena populations. Can these (and other?) regions be indicated in Figure S1? Did the nuclear markers reveal anything?

Answer: Figure S1 has been completed with an Andalusian map showing the Guadalquivir Valley, Betic Cordillera and Northern Sierra Morena adapted from Fernández i Marti et al., 2018 Forests 2018, 9, 337. Moreover, we have located the Andalusian populations used in all our works. Regarding nuclear markers, “Our nuclear DNA showed a much weaker spatial structure (differentiation among populations) than the chloroplast DNA.” (This information has been clarified in the revised version of the manuscript). This is deeply discussed in our previous publication (Fernández i Marti et al., 2018 Forests 2018, 9, 337)

Line 472/473/474. Please rephrase this sentence as it is now not clear. Maybe DELETE “, when the patterns are similar for orthodox and recalcitrant seeds”?

Answer: It has been clarified in the revised version.

Supplementary Tables: I appreciate the work the authors have undertaken to include results. I am sure future readers will appreciate this as well!

Figure S1: I suggest adding the names of the various regions, especially those that are mentioned in the text.

Answer: All the Andalusian population used in our works has been included in the Figure S1.

Text change suggestions:

Line 120/121: Please clarify what is said here, I seem to misinterpret this sentence. 4.5 MHa is more than 20% of 15 MHa.

Answer: Thank you for this comment. It has been revised. It was 30 %.

Line 156: REPLACE “in the main cause” WITH “is the main cause”

Answer: Done

Line 175: REPLACE “system (Table 1[8]).” WITH “system (Table 1[8]). This includes the high variability found in natural populations of Q. ilex, while Populus and Eucalyptus studies can use clones with sequenced genomes.”  (NOTE that I used here the comment you gave in your response).

Answer: Done

Line 182: DELETE “relevant”

Answer: Done

Line 232: REPLACE “and -omics techniques are presented.” WITH “and -omics techniques.”

Answer: Done

Line 240: DELETE “, for use as an isozyme”

Answer: Done

Line 249/250: REPLACE “exploring the distribution of differentiation at the genome level in” WITH “differentiating between” [Check if this still captures what you want to say here]

Answer: Done

Line 256: REPLACE “, more noticeable in” WITH “, especially in”

Answer: Done

Line 264: REPLACE “neing” with “being”             

Answer: Done

Line 282: REPLACE “10 chloroplasts that” WITH “10 chloroplast SSRs that”

Answer: Done

Line 282: REPLACE “that indicate maternal” WITH “ to examine maternal”

Answer: Done

Line 283: REPLACE “that indicate chromosomal” WITH “to examine chromosomal”

Answer: Done

Line 284: REPLACE “found with” WITH “found for”

Answer: Done

Line 370: REPLACE “described in” WITH “ described for”

Answer: Done

Line 373: REPLACE “reported in” WITH “reported for”

Answer: Done

Line 374: REPLACE “with 190 loci represented, 69 %” WITH “for 190 loci: 69%”

Answer: Done

Line 399: REPLACE “For this, one” WITH “One”

Answer: Done

Line 411: DELETE “is”

Answer: Done

Line 412: “is the responses to”  EITHER “is the response to” OR “are the responses to”

Answer: Done

Line 414: DELETE “the”

Answer: Done

Line 426: REPLACE “in to mRNA profile of 28 % of” WITH “to 28% of”

Answer: Done

Line 472: REPLACE “concluded” with “indicate”

Answer: Done

Line 488: REPLACE “approached by using” WITH “examined by”

Answer: Done

Line 512: italicize “Q. ilex”

Answer: Done

Line 525/526 REWRITE TO IMPROVE CLARITY please

Answer: This sentence has been clarified.

Line 532: REPLACE “of potential use in breeding programs based on the selection of elite tolerant genotypes.” WITH “to be used for the selection of elite tolerant genotypes in a breeding program.”

Answer: Done

Line 553 etc: REPLACE “As a result, the following variable proteins are proposed as putative markers for resilience in Q. ilex, namely, aldehyde dehydrogenase, glucose-6-phosphate isomerase, 50S ribosomal protein L5, and α-1,4-glucan-protein synthase (UDP-forming).” WITH “As a result, aldehyde dehydrogenase, glucose-6-phosphate isomerase, 50S ribosomal protein L5, and α-1,4-glucan-protein synthase (UDP-forming) are proposed as putative markers for resilience in Q. ilex.”

Answer: Done

Line 603: REPLACE “is generating” WITH “has generated”

Answer: Done

Line 723: REPLACE “Research” WITH “ research”

Answer: Done

Line 765: REPLACE “Benthamiana” with “benthamiana”

Answer: Done

Line 766: REPLACE “in” with “for”

Answer: Done